# STEPFORWARD study: a randomised controlled feasibility trial of a self-aligning prosthetic ankle-foot for older patients with vascular-related amputations

Natalie Vanicek [ID],[1] Elizabeth Coleman,[2] Judith Watson [ID],[2] Kerry Bell,[2] Catriona McDaid [ID],[2] Cleveland Barnett,[3] Martin Twiste,[4] Fergus Jepson,[5] Abayomi Salawu [ID],[6] Dennis Harrison,[7] Natasha Mitchell[2]

For numbered affiliations see end of article.

**Correspondence to**
Professor Natalie Vanicek;
n.vanicek@hull.ac.uk

## ABSTRACT

**Objectives** To determine the feasibility of conducting a full-scale randomised controlled trial (RCT) of the effectiveness and cost-effectiveness of a self-aligning prosthetic ankle-foot compared with a standard prosthetic ankle-foot.

**Design** Multicentre parallel group feasibility RCT.

**Setting** Five prosthetics centres in England recruiting from July 2018 to August 2019.

**Participants** Adults aged ≥50 years with a vascular-related or non-traumatic transtibial amputation for 1 year or longer, categorised as having 'limited community mobility' and using a non-self-aligning ankle-foot.

**Intervention** Participants were randomised into one of two groups for 12 weeks: self-aligning prosthetic ankle-foot or existing non-self-aligning prosthetic ankle-foot.

**Outcomes** Feasibility measures: recruitment, consent and retention rates; and completeness of questionnaire and clinical assessment datasets across multiple time points. Feasibility of collecting daily activity data with wearable technology and health resource use data with a bespoke questionnaire.

**Results** Fifty-five participants were randomised (61% of the target 90 participants): n=27 self-aligning ankle-foot group, n=28 non-self-aligning ankle-foot group. Fifty-one participants were included in the final analysis (71% of the target number of participants). The consent rate and retention at final follow-up were 86% and 93%, respectively. The average recruitment rate was 1.25 participants/site/month (95% CI 0.39 to 2.1). Completeness of questionnaires ranged from 89%–94%, and clinical assessments were 92%–95%, including the activity monitor data. The average completion rates for the EQ-5D-5L and bespoke resource use questionnaire were 93% and 63%, respectively.

**Conclusions** This feasibility trial recruited and retained participants who were categorised as having 'limited community mobility' following a transtibial amputation. The high retention rate of 93% indicated the trial was acceptable to participants and feasible to deliver as a full-scale RCT. The findings support a future, fully powered evaluation of the effectiveness and cost-effectiveness

## Strengths and limitations of this study

► This was one of few clinical trials investigating ankle-foot prostheses within prosthetics services in the UK. The study design is a multicentre, randomised controlled trial in an under-researched clinical population involving prosthetics.
► The completeness of subjective and objective assessment measures, administered in a clinical setting and via post, was high.
► Feasibility and acceptability were ascertained with quantitative and trial procedure data.
► Recruitment was lower and slower than anticipated, but the ability to retain participants was high.
► Effectiveness and cost-effectiveness of a self-aligning ankle-foot will need to be established in a definitive full-scale trial.

of a self-aligning prosthetic ankle-foot compared with a standard non-self-aligning version with some adjustments to the trial design and delivery.

**Trial registration number** ISRCTN15043643.

## INTRODUCTION

In the UK, there are approximately 6000 new referrals to prosthetics services every year. Most of these referrals are patients aged over 50 years who have had an amputation at the transtibial (below-knee) level.[1] The majority of lower limb amputations result from long-term diabetes mellitus, coronary and peripheral vascular diseases.[2 3] Consequently, most new referrals for a prosthesis involve older adults who have other health comorbidities and who are categorised as having 'limited community mobility'. These patients are an under-researched group, with few studies focused on their mobility with a prosthesis.

Many factors determine the type of prosthesis that a patient receives, such as estimation of patient outcomes and goals, and the healthcare budget.[4] A Cochrane review reported that there was insufficient evidence from robust studies to support the 'overall superiority of any individual type of prosthetic ankle-foot mechanism'.[5] There are also no standardised criteria for the prescription of ankle-foot prostheses. Practice varies across UK centres and is frequently cost driven. Subsequently, the majority of older people with a transtibial amputation receive a standard, non-self-aligning ankle-foot, such as the non-articulated solid ankle cushioned heel (SACH), uniaxial or multiaxial prosthetic foot. These prosthetic ankle-feet have limited functionality. The SACH foot has a solid 'ankle' that does not allow any movement. The uniaxial ankle-foot is only capable of moving about a single axis with some ankle dorsi- and plantar- flexion; the multiaxial ankle-foot can rotate about multiple axes, allowing ankle dorsi- and plantar- flexion and eversion–inversion. The elastic recoil to return to a plantigrade position makes the uniaxial and multiaxial ankle-feet unable to self-align to inclined surfaces, making these feet more suitable for standing or walking on level ground. This requires the user to make compensatory actions at proximal joints (compensations at the hip bilaterally and intact knee of the affected limb) and at the intact ankle to achieve stability with a 'foot flat' position when walking on slopes.[6 7] These actions result in greater asymmetry between the two limbs and increase the metabolic requirements of walking.[6] This can make walking more difficult and tiring. Our public involvement members reported that a poorly functioning prosthesis can contribute to sedentary behaviour, pain and more frequent visits to prosthetics centres and other healthcare services, which may lead to disuse and poorer quality of life. Therefore, identifying a more functionally suitable prosthesis could benefit patients in meaningful ways following amputation.

A 'family' of prosthetic ankle-feet that use a hydraulic mechanism to self-align to sloped surfaces have been designed to attenuate many of the burdens associated with standard, non-self-aligning ankle-feet. These prosthetic ankle-feet have been designed for users of varying ability (more active users, K3–K4 classification) and also for users categorised as having 'limited community mobility' (K2 users).[8] The hydraulic mechanism allows the prosthetic ankle-foot to self-align to sloped surfaces (due to ground reaction forces) and remain in a dorsiflexed position throughout the swing phase. This can improve ground clearance and secure the biological knee, which is important for falls prevention.[9] Previous laboratory-based studies reported that a self-aligning ankle-foot reduced residuum-socket interface pressures in active users,[10] which could alleviate pain in the residuum longer term, improved walking[11] and enhanced quality of life[12 13] in users with lower activity levels.

More functional prostheses may improve patient function and mobility following a lower limb amputation.[14] However, the effects of a self-aligning prosthesis have not been evaluated with robust randomised controlled trials (RCTs) in a community setting. In particular, there is a lack of research involving participants who have had a non-traumatic (eg, vascular-related) amputation. A self-aligning ankle-foot is more expensive than a non-self-aligning version by approximately £900 for the K2-user version (at 2020 UK prices) with variations in price across manufacturers. However, the potential patient benefits, such as improved mobility, better quality of life and reduced falls, could offset future healthcare and socio-economic costs.

There is a need for an RCT of the effectiveness and cost-effectiveness of a self-aligning prosthetic ankle-foot for older patients, with a vascular-related or non-traumatic transtibial amputation, compared with a standard, non-self-aligning ankle-foot. The STEPFORWARD study used a mixed-methods approach to determine feasibility of conducting such a study. A feasibility study was considered important in advance of a full-scale RCT given the lack of large community-based RCTs in this field. A full-scale trial will be powered appropriately to detect a difference in the primary outcome between the two trial arms (and as such has a sample size large enough to answer a research question on intervention effectiveness). A feasibility study addresses the question of whether a specific study can be done, whether it should proceed and, if so, how.[15] It is not designed to address the question of whether an intervention works. This article reports the feasibility as determined by the quantitative data, which was assessed according to: (1) participant recruitment, including time to recruit, consent and retention rates and adverse events (AEs); (2) completeness of data across multiple time points, including health economics data while piloting a bespoke health resource use questionnaire and (3) day-to-day activity measured with wearable technology. The findings from this work could be relevant to clinical practitioners, researchers and policy-makers.

## METHODS

### Study design and participants

The STEPFORWARD trial was a multicentre, parallel group, randomised controlled feasibility trial to assess the possibility of conducting a full-scale RCT in the future, which would be adequately powered to answer the primary research question. The full study protocol, with detailed inclusion/exclusion criteria and recruitment/screening pathways, has been published[16] and is briefly described below.

We aimed to recruit 90 participants who were aged ≥50 years with a unilateral transtibial amputation for ≥1 year due to vascular, neurological or life-limiting reasons, who were categorised as having 'limited community mobility', with a stable residuum, and using a non-self-aligning prosthetic ankle-foot (eg, SACH, uniaxial, multiaxial (eg, multiflex) or other K1/K2 feet). We excluded anyone who had a contraindication to wearing their current prosthesis or the self-aligning ankle-foot (according to

the manufacturer's specifications), had experienced a recent cerebrovascular event or had a disease affecting their memory. Allowing for 20% attrition, we anticipated 72 participants in our final analysis.

## Recruitment and randomisation

We used a two-stage screening process to identify eligible patients; full details are provided in the published protocol.[16] In summary, potential participants were identified by database screening or during routine appointments to five prosthetics centres in England from July 2018 to August 2019. They were posted a study invitation pack and asked to return a Consent to Contact form if they were interested in being contacted about the trial. A member of their multidisciplinary team then contacted them and completed the first section of the Screening Form over the telephone. If potentially eligible, they were invited into clinic to complete the outstanding screening questions, which had to be done face to face, and provide their written consent. Following consent, eligible patients completed all baseline measures before being randomised into one of two trial arms: keep their non-self-aligning prosthetic ankle-foot (standard treatment group) or receive a new self-aligning ankle-foot (intervention group). Participants were randomised individually, stratified according to prosthetics centre on a 1:1 basis, by a telephone randomisation service set up by York Trials Unit. Blinding of the participants and investigators was not possible because of the nature of the intervention.

## Standard treatment and intervention

Participants in the standard treatment group continued using their normal prosthetic ankle-foot and had access to all clinical services as normal. They were asked to carry on with their normal daily routine during the intervention period.

Participants in the intervention group were fitted with the self-aligning prosthetic ankle-foot by their regular prosthetist. The self-aligning ankle-foot used in this trial was the Avalon^K2, manufactured by Blatchford, Basingstoke, UK (Patent reg: 5336386), which was already commercially available and could be prescribed under the national health Service (NHS). Once fitted, participants were asked to ambulate with it, as they would normally with any new prosthetic ankle-foot for approximately 12 weeks after fitting (intervention period). At the end of the study, participants in the intervention group only could keep the self-aligning ankle-foot if they preferred it. Participants in standard treatment group were not offered the self-aligning ankle-foot at the end of the study as it is not known whether it is better or more acceptable to users than the standard prosthesis.

## Outcome measures

The primary outcome of this feasibility study related to collecting data on recruitment, consent and retention rates. Reasons for non-participation or withdrawal were recorded where possible. We assessed the completeness of data across multiple time points: baseline, interim and final follow-ups. The baseline and final assessments were conducted at the participant's usual prosthetics centre. At the interim follow-up, only questionnaire data were collected via post. Data were gathered from questionnaires (baseline demographics and bespoke health resource use questionnaires, the Locomotor Capabilities Index (LCI-5),[17] Houghton Scale,[18] the Patient-Reported Outcomes Measurement Information System (PROMIS) Short-Form V.1.0 Pain (3a and 8a) questionnaires[19] and EQ-5D-5L[20] and from clinical assessments (Timed Up and Go (TUG) test,[21] Timed Up and Down Stairs (TUDS) test,[22] 2 min walk test (2mWT)[23] and Berg Balance Scale (BBS).[24] Data related to daily step count and time spent walking were collected from a loaned activity monitor (activPAL4, PAL Technologies, Glasgow, UK) worn on the prosthesis for 1 week. At baseline, all participants wore their normal (non-self-aligning) prosthetic ankle-foot during the clinical assessments and activity monitor data collection; at final follow-up they wore whichever prosthetic ankle-foot they were using currently. AEs related to the prosthesis only, and any serious AEs (SAEs), were recorded throughout the study.

## Statistical analyses

The recruitment rate was reported monthly, and overall, by site. An average monthly recruitment rate was calculated, with a 95% CI estimated from the data collected. The number of eligible patients and those approached for consent was summarised overall, by site, using counts and percentages.

Baseline data were summarised by trial arm, as randomised, with no formal comparison between the groups. Continuous data were reported descriptively and categorical data by counts and percentages. Completion rates of all the clinical outcome measures were reported by trial arm and overall.

## Health economic analysis

The health economics work investigated the feasibility of collecting self-reported patient use of health service resources (eg, primary care visits, prescription medication) over the study period and to evaluate the appropriateness of these data. This information would help to refine the bespoke questionnaire in a future trial. A full cost-effectiveness analysis was not completed as this was a feasibility study. Nonetheless, the costing approach was undertaken from an NHS perspective; unit costs were derived from established national costing sources such as NHS Reference Costs[25] and Personal Social Services Research Unit costs of health and social care.[26]

## RESULTS
### Feasibility assessment
#### Recruitment

Over 880 potential participants were initially screened by searching patient databases across the five prosthetics

centres. Seventy-eight were identified as potentially eligible after the first screening, but 12 were subsequently excluded following the final, second screening. Eighty-six per cent of those screened, who were found to be fully eligible, consented to participating (55 of 64 participants) in the trial. The denominator was 64 as two were randomised in error and subsequently withdrawn (figure 1). As we used a two-stage screening process, only potential participants who met the initial inclusion criteria were approached; those uninterested in being involved during the initial screening were not screened further. Therefore, the consent rate may be artificially high. Reasons for non-participation are also presented in figure 1. In total, 55 participants (61% of the target 90 participants) were included and randomised as follows: 27 to the self-aligning prosthetic ankle-foot (intervention) group and 28 to the non-self-aligning (standard treatment) group.

Initial recruitment was slower than anticipated, therefore, two additional recruiting sites were included midway through the trial, bringing the total number of participating sites to five. The average recruitment rate was 1.25 participants/site/month (95% CI 0.39 to 2.1) ranging from 0.83 to 2.2 participants per month. Each site recruited approximately the same number of participants (mean=11, range=9–13 participants), despite them being open to recruitment for different lengths of time (range=5.0–13.7 months).

### Participants
The mean (SD) age of participants in this study was 68.8 (9.6) years, ranging from 51.8 to 86.8 years. Most participants were male (85.5%) and all classified themselves as White British, Irish or other. The most common reason for amputation was diabetes (45.5%). Full participant characteristics are described in table 1 .

### Follow-up and AEs
In addition to the two people randomised in error and withdrawn, a further two participants who were allocated to the intervention group withdrew from the study: one as a result of having suffered a stroke and one reported the self-aligning prosthetic ankle-foot was too heavy. At final follow-up, 51 of the 55 participants returned the questionnaires giving a retention rate of 93%. This was 71% of the planned target of 72 participants included in the final analysis.

A total of four SAEs and one AE were reported. The SAEs were due to the following reasons: stroke; fall and femoral fracture when not wearing their prosthesis (both of these cases occurred in the intervention group); fall and hip fracture, leading to hospitalisation and subsequently death; and recurrent falls with fibular fracture (both of these cases were in the standard treatment group). The AE occurred because one participant (intervention group) developed blistering on their residuum.

### Data completeness
There were four standardised measures completed at all three time points: LCI-5, Houghton, PROMIS 3a and PROMIS 8a. The average completion rate across all time points ranged from 89% to 94%. Data from four clinical assessments (TUG, TUDS, 2mWT and BBS), and an activity monitor worn for a week, were collected at both baseline and final follow-up, and their average completion rate ranged from 92% to 95%. All the measures were well completed, or attempted, with at least 75% completed in full. Details of the outcome measures and completion rates can be found in tables 2 and 3.

None of the four standardised measures appeared to show a change between baseline and follow-up. For the 2mWT, participants with the self-aligning prosthetic ankle-foot walked on average further at final follow-up (mean (SD): 88.8 (40.0) m) than at baseline (81.0 (31.4) m). This equates to an average increase of 6.2 (16.2) m in walking distance for the participants in the intervention group between these two time points. However, in the standard treatment group, there was an average decrease of 9.0 (29.8) m, indicating that these participants were walking less distance over 2 min at the final follow-up. Scores for the TUG, TUDS and BBS remained largely unchanged across both groups. Activity monitor data revealed that the daily time spent stepping between baseline and follow-up was mostly unchanged for both groups, which was approximately 30 min per day. Conversely, the number of steps taken per day decreased in the intervention group (1755.5 (1963.7) steps at baseline vs 1673.2 (1594.4) steps at final follow-up) yet increased in the standard treatment group (1750.9 (1369.9) steps at baseline vs 1836.1 (1647.8) steps at final follow-up). The average change in number of steps taken per day decreased by 2.9 (756.2) steps in the intervention group between baseline and final follow-up, yet increased by 283.3 (1614.8) steps in the standard treatment group. In both groups, the variability in daily stepping was high.

### Health economic data
The health economic outcomes are based on the patient-completed questionnaires, which incorporated the EQ-5D-5L and a bespoke resource use questionnaire. The average completion rates were 93% and 63%, respectively. Over 89% of participants completed the EQ-5D-5L fully. At baseline, responses to the EQ-5D-5L indicated relatively equal levels of health utility between the trial arms. By final follow-up, an increase in mean utility of 0.13 in the intervention group was observed, while mean utility in the standard treatment group remained the same. A positive mean difference in favour of the self-aligning prosthetic ankle-foot in the intervention group remained consistent after adjusting for baseline utility (table 4).

For the resource use questionnaire, the completion rates did not differ by the respective components, suggesting all sections were acceptable and each type of data was feasible to collect. Resource use was minimal across both primary and secondary/tertiary care for participants in

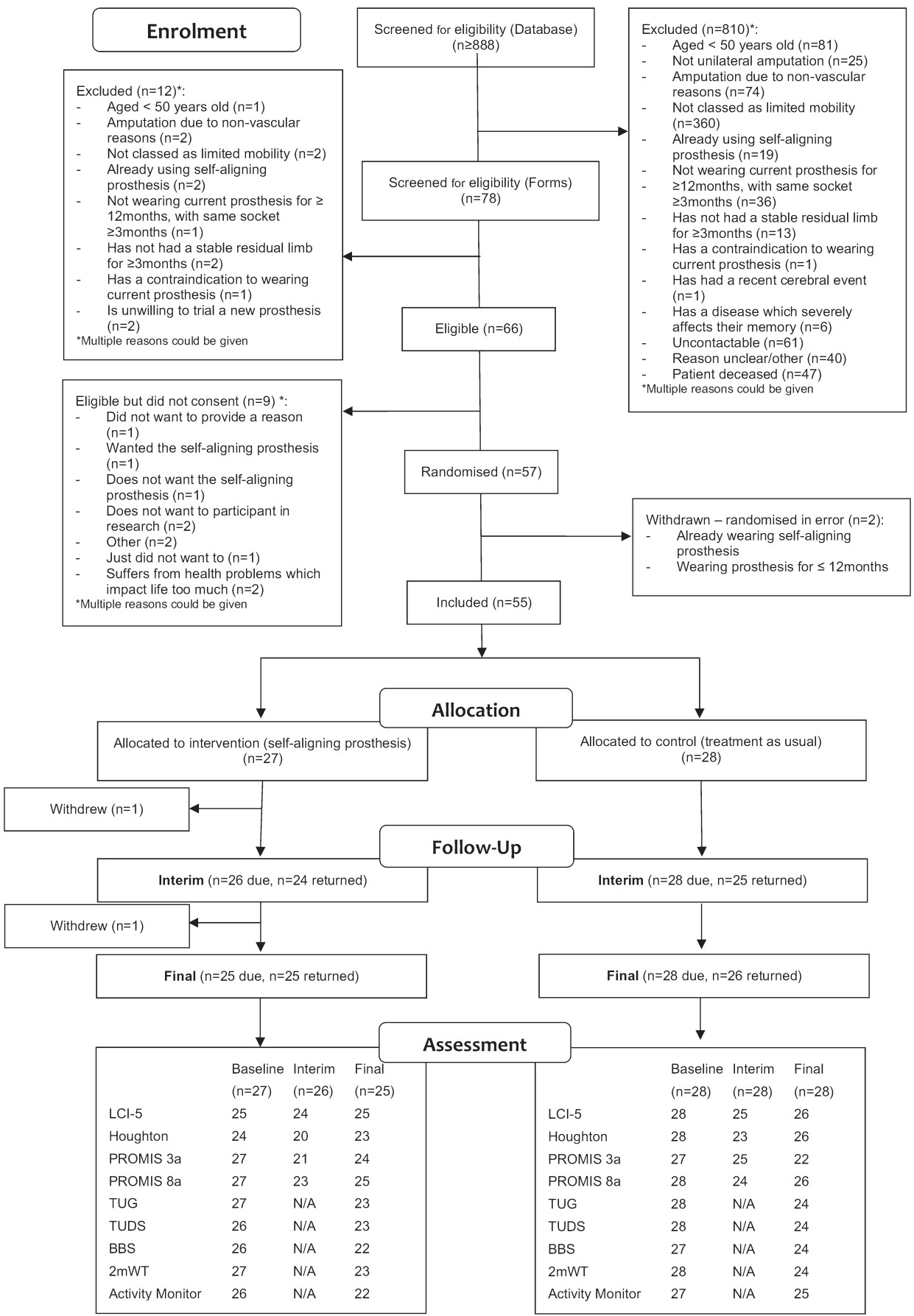

**Figure 1** Consort flow chart for the STEPFORWARD study. LCI-5, Locomotor Capabilities Index; PROMIS, Patient-Reported Outcomes Measurement Information System; TUDS, Timed Up and Down Stairs; TUG, Timed Up and Go.

**Table 1** Demographic characteristics according to trial arm

| | Self-aligning (n=27) | Standard (n=28) | Overall (n=55) |
|---|---|---|---|
| **Age, years** | n=27 | n=28 | n=55 |
| Mean (SD) | 70.0 (9.2) | 67.6 (10.0) | 68.8 (9.6) |
| Median (min, max) | 71.5 (52.7, 86.8) | 68.3 (51.8, 84.5) | 70.7 (51.8, 86.8) |
| **Gender, n(%)** | | | |
| Male | 23 (85.2) | 24 (85.7) | 47 (85.5) |
| Female | 4 (14.8) | 4 (14.3) | 8 (14.6) |
| **Ethnicity, n(%)** | | | |
| White (British, Irish or Other) | 27 (100.0) | 28 (100.0) | 55 (100.0) |
| Black (African, Caribbean or Other) | 0 (0.0) | 0 (0.0) | 0 (0.0) |
| Asian (Indian, Pakistani, Bangladeshi or Other) | 0 (0.0) | 0 (0.0) | 0 (0.0) |
| Mixed Background | 0 (0.0) | 0 (0.0) | 0 (0.0) |
| Chinese | 0 (0.0) | 0 (0.0) | 0 (0.0) |
| **Marital status, n(%)** | | | |
| Living alone never married | 2 (7.4) | 3 (10.7) | 5 (9.1) |
| Living with partner | 4 (14.8) | 0 (0.0) | 4 (7.3) |
| Married/civil partnership | 9 (33.3) | 19 (67.9) | 28 (50.9) |
| Separated | 2 (7.4) | 1 (3.6) | 3 (5.5) |
| Divorced | 5 (18.5) | 4 (14.3) | 9 (16.4) |
| Widowed | 2 (7.4) | 1 (3.6) | 3 (5.5) |
| Missing | 3 (11.1) | 0 (0.0) | 3 (5.5) |
| **Main activity, n(%)** | | | |
| Full-time employment | 1 (3.7) | 3 (10.7) | 4 (7.3) |
| Part-time employment | 0 (0.0) | 1 (3.6) | 1 (1.8) |
| Self-employed | 0 (0.0) | 0 (0.0) | 0 (0.0) |
| Unable to work due to poor health | 1 (3.7) | 5 (17.9) | 6 (10.9) |
| Unemployed | 0 (0.0) | 1 (3.6) | 1 (1.8) |
| Retired | 24 (88.9) | 17 (60.7) | 41 (74.6) |
| Other | 1 (3.7) | 1 (3.6) | 2 (3.6) |
| **Smoking status, n(%)** | | | |
| Ex-smoker | 15 (55.6) | 17 (60.7) | 32 (58.2) |
| Current smoker | 1 (3.7) | 3 (10.7) | 4 (7.3) |
| Never smoked | 11 (40.7) | 8 (28.6) | 19 (34.6) |
| **Diabetes status, n(%)** | | | |
| Yes | 19 (70.4) | 18 (64.3) | 37 (67.3) |
| No | 8 (29.6) | 10 (35.7) | 18 (32.7) |
| **If yes:** | | | |
| Type one | 5 (26.3) | 2 (11.1) | 7 (18.9) |
| Type two | 12 (63.2) | 13 (72.2) | 25 (67.8) |
| Missing | 2 (10.5) | 3 (16.7) | 5 (13.5) |
| **Cause of amputation, n(%)[a]** | | | |
| Diabetes | 14 (51.9) | 11 (39.3) | 25 (45.5) |
| Peripheral vascular disease | 10 (37.0) | 10 (35.7) | 20 (36.4) |
| Blood clot | 3 (11.1) | 2 (7.1) | 5 (9.1) |
| Deep vein thrombosis | 1 (3.7) | 0 (0.0) | 1 (1.8) |
| Aneurysm | 0 (0.0) | 1 (3.6) | 1 (1.8) |

Continued

**Table 1** Continued

|  | Self-aligning (n=27) | Standard (n=28) | Overall (n=55) |
|---|---|---|---|
| Diabetic neuropathy | 2 (7.4) | 1 (3.6) | 3 (5.5) |
| Other neuropathy | 1 (3.7) | 0 (0.0) | 1 (1.8) |
| Cancer | 1 (3.7) | 1 (3.6) | 2 (3.6) |
| Other | 5 (18.5) | 6 (21.4) | 11 (20.0) |

both arms of the trial. The most frequently received care was via outpatient attendances at NHS hospitals (mean (SD): 0.21 (0.51) rising to 2.04 (3.72) visits from baseline to final follow-up for the intervention group; 0.08 (0.27) rising to 1.14 (2.19) visits for the standard treatment group.

Medications were grouped according to comorbidity to investigate their inclusion in a full-scale trial as follows: blood pressure/heart problems, neuropathic pain, diabetes, phantom pain. Participant uptake of the groups of medications explored using the resource use questionnaire was high, and there were some differences between groups, suggesting a micro-costing exercise including medications data may be merited in a full-scale trial. Approximately 73% of participants overall reported taking medications related to blood pressure and/or heart problems, and 62% of participants overall took medications for diabetes. Medication use appeared to increase over the course of the trial for participants in both arms.

The intervention (self-aligning prosthetic ankle-foot) was delivered in the NHS setting by the usual staff providing post-amputation aftercare (prosthetist, physiotherapist). Other than the self-aligning ankle-foot itself, participants received equal care to standard treatment. The cost (at 2020 UK prices) of the self-aligning ankle-foot used in the trial was £1192 compared with the non-self-aligning version estimated at £287 (although this depends on the exact standard ankle-foot and would need to be determined accurately based on the user's actual foot). The additional intervention cost to the NHS of delivering the self-aligning ankle-foot is therefore approximately £905.

### Identification of primary outcome measure

The findings from this study have demonstrated feasibility, acceptability and a signal of efficacy for three measures: walking ability with the 2mWT, daily activity with activity monitor data and quality of life with the EQ-5D-5L. We consulted with our patient advisory group (PAG) to inform the appropriateness of a primary outcome to be used in a future full-scale trial. The PAG members reported that, of the three measures, the EQ-5D-5L resonated most strongly with them as it captured the constructs that were important to them (eg, mobility, pain, usual activities, general health). This measure will be considered for use as the primary outcome in a future trial.

## DISCUSSION
### Main findings

The primary objective of this study was to determine the feasibility of conducting a future, full-scale RCT of the effectiveness and cost-effectiveness of a self-aligning prosthetic ankle-foot compared with a standard prosthetic ankle-foot. There are few clinical trials involving older adults living with a lower limb amputation, who are an under-researched but growing group of patients. The consent, retention and completion rates were high, demonstrating that it is feasible to recruit and retain participants to a future trial addressing the research question. While the consent rate may be artificially high, due to screening methods used in the current study, the final retention and completion rates indicate the study was acceptable to participants.

We did not meet the recruitment target of 90. Given that only 7.4% of screened patients met the eligibility criteria (though only 7.2% were truly eligible, due to the randomisation in error), it is important to consider whether the criteria were too restrictive and should be adjusted in a future trial. It is common that the residuum undergoes considerable volume changes within the first year post-amputation, necessitating the fabrication of new prosthetic sockets. As the intervention period of this feasibility study was only 12 weeks long (due to resource constraints), we sought to recruit established prosthesis users who had no current contraindications to wearing their ankle-foot prosthesis, had a stable residuum volume and the same prosthetic socket for ≥3 months and were free of open wounds or infections on the residuum. We also only included patients who had a transtibial amputation secondary to a health condition, as this group is most likely categorised as having 'limited community mobility' and frequently prescribed a non-self-aligning prosthetic ankle-foot. It would be clinically relevant to broaden the eligibility criteria to include any patient with a transtibial amputation within this mobility category, regardless of cause and time since amputation, in a future study. This would widen the pool of patients potentially eligible for the trial. However, we do not have data from the feasibility study to assess impact on recruitment and it would be important for the future trial to have an internal pilot with clear stop-go criteria to confirm the feasibility of a modified study design.

Five AEs, of which four were categorised as serious, occurred throughout this trial. All of the SAEs (one

**Table 2** Standard outcomes at each time point (baseline, interim and final follow-up), by trial arm and overall

| | Self-aligning (n=27) | Standard (n=28) | Overall (n=55) |
|---|---|---|---|
| **LCI-5 overall score (0–56): average completion 94%** | | | |
| **Baseline** | N=25 | N=28 | N=53 |
| Mean (SD) | 36.4 (12.0) | 39.6 (11.2) | 39.1 (11.6) |
| Median (p25, p75) | 38 (30.3, 46) | 38 (31, 50) | 38 (30.3, 47) |
| **Interim** | N=24 | N=25 | N=49 |
| Mean (SD) | 37.3 (12.3) | 38.5 (10.8) | 37.9 (11.4) |
| Median (p25, p75) | 39.5 (28.5, 47.5) | 41 (30, 44.7) | 41 (29, 46) |
| **Final** | N=25 | N=26 | N=51 |
| Mean (SD) | 37.8 (11.4) | 39.2 (11.2) | 38.5 (11.2) |
| Median (p25, p75) | 40 (30, 47) | 40.5 (29, 49) | 40 (29, 47) |
| **Houghton overall score (0–12): average completion: 89%** | | | |
| **Baseline** | N=24 | N=28 | N=52 |
| Mean (SD) | 7.8 (2.0) | 8.8 (1.8) | 8.3 (1.9) |
| Median (p25, p75) | 8 (7.5, 9) | 9 (8, 10) | 9 (8, 9) |
| **Interim** | N=20 | N=23 | N=43 |
| Mean (SD) | 8.4 (2.0) | 8.5 (1.3) | 8.5 (1.6) |
| Median (p25, p75) | 9 (7, 10) | 9 (8, 9) | 9 (8, 10) |
| **Final** | N=23 | N=26 | N=49 |
| Mean (SD) | 8.3 (1.9) | 8.5 (2.0) | 8.4 (1.9) |
| Median (p25, p75) | 8 (8, 10) | 9 (8, 9) | 9 (8, 9) |
| **PROMIS 3a overall score (30.7–71.8): average completion: 90%** | | | |
| **Baseline** | N=27 | N=27 | N=54 |
| Mean (SD) | 41.3 (10.0) | 43.1 (9.5) | 42.2 (9.7) |
| Median (p25, p75) | 43.5 (30.7, 52.1) | 43.5 (30.7, 49.4) | 43.5 (30.7, 49.4) |
| **Interim** | N=21 | N=25 | N=46 |
| Mean (SD) | 43.6 (9.6) | 45.6 (10.3) | 44.7 (10.0) |
| Median (p25, p75) | 43.5 (30.7, 49.4) | 46.3 (40.2, 52.1) | 46.3 (30.7, 52.1) |
| **Final** | N=24 | N=22 | N=46 |
| Mean (SD) | 40.4 (9.4) | 42.7 (8.7) | 41.4 (9.0) |
| Median (p25, p75) | 40.2 (30.7, 47.9) | 44.9 (30.7, 49.4) | 43.5 (30.7, 49.4) |
| **PROMIS 8a overall score (40.7–77.0): average completion: 94%** | | | |
| **Baseline** | N=27 | N=28 | N=55 |
| Mean (SD) | 52.8 (11.2) | 51.7 (9.4) | 52.2 (10.2) |
| Median (p25, p75) | 55 (40.7, 63.5) | 51.8 (40.7, 58.5) | 52.3 (40.7, 60.2) |
| **Interim** | N=23 | N=24 | N=47 |
| Mean (SD) | 53.5 (11.3) | 56.7 (8.9) | 55.1 (10.1) |
| Median (p25, p75) | 55 (40.7, 64.1) | 58.8 (52.3, 63.2) | 57.4 (40.7, 63.5) |
| **Final** | N=25 | N=26 | N=51 |
| Mean (SD) | 51.2 (11.4) | 53.3 (10.0) | 52.3 (10.7) |
| Median (p25, p75) | 49.9 (40.7, 60.2) | 54.5 (40.7, 60.2) | 53.2 (40.7, 60.2) |

LCI-5: higher scores indicate a greater locomotor capability with the prosthesis; Houghton: higher scores indicate greater performance and comfort; PROMIS 3a (pain intensity): coverts raw scores to T-scores, higher scores indicate more intense pain; PROMIS 8a: higher scores indicate more pain interference.

LCI-5, Locomotor Capabilities Index; PROMIS, Patient-Reported Outcomes Measurement Information System.

**Table 3** Clinical assessments (TUG, TUDS, 2mWT and BBS) at baseline and final follow-up, by trial arm and overall

| | Self-aligning (n=27) | Standard (n=28) | Overall (n=55) |
|---|---|---|---|
| **TUG, seconds: average completion 95%** | | | |
| **Baseline** | N=27 | N=28 | N=55 |
| Mean (SD) | 20.8 (15.2) | 21.3 (23.5) | 21.1 (19.7) |
| Median (p25, p75) | 17 (12, 24) | 14.5 (11.5, 23) | 17 (12, 23) |
| Unable to complete | 0 | 0 | 0 |
| **Final** | N=23 | N=22 | N=45 |
| Mean (SD) | 21.0 (19.1) | 17.8 (8.4) | 19.5 (14.8) |
| Median (p25, p75) | 16 (11, 13) | 14.5 (13, 22) | 15 (11, 23) |
| Unable to complete | 0 | 2 | 0 |
| **TUDS, seconds: average completion 93%** | | | |
| **Baseline** | N=17 | N=27 | N=44 |
| Mean (SD) | 61.4 (33.0) | 66.9 (38.5) | 64.7 (36.2) |
| Median (p25, p75) | 50 (39, 72) | 51 (41, 88) | 50.5 (41, 88) |
| Unable to complete | 9 | 1 | 10 |
| **Final** | N=19 | N=21 | N=40 |
| Mean (SD) | 59.2 (29.2) | 68.1 (56.7) | 63.9 (45.4) |
| Median (p25, p75) | 51 (33, 79) | 48 (39, 74) | 48.5 (38, 76.5) |
| Unable to complete: | 4 | 3 | 7 |
| **2mWT, metres: average completion 95%** | | | |
| **Baseline** | N=27 | N=28 | N=55 |
| Mean (SD) | 81.0 (31.4) | 94.7 (33.5) | 87.9 (33.5) |
| Median (p25, p75) | 77 (60, 110) | 95 (68, 113) | 83 (61, 113) |
| **Final** | N=23 | N=24 | N=47 |
| Mean (SD) | 88.8 (40.0) | 83.9 (41.5) | 86.3 (39.5) |
| Median (p25, p75) | 80 (62, 120) | 80 (57.5, 106) | 80 (60, 108) |
| **BBS, scored 0–56: average completion 92%** | | | |
| **Baseline** | N=26 | N=27 | N=53 |
| Mean (SD) | 39.5 (11.3) | 42.9 (8.1) | 41.3 (9.8) |
| Median (p25, p75) | 41.5 (32, 48) | 43 (36, 51) | 43 (34, 48) |
| **Final** | N=22 | N=24 | N=46 |
| Mean (SD) | 40.7 (12.9) | 39.4 (11.6) | 40.0 (12.1) |
| Median (p25, p75) | 44 (30, 51) | 41.5 (33, 49) | 41.5 (33, 50) |
| **Activity monitor: minutes stepping per day** | | | |
| **Baseline** | N=26 | N=27 | N=53 |
| Mean (SD) | 29.9 (30.8) | 31.1 (24.9) | 30.5 (27.7) |
| Median (p25, p75) | 24.7 (9.4, 31.4) | 26 (14.5, 40.3) | 25.9 (10.6, 34.1) |
| **Final** | N=22 | N=25 | N=47 |
| Mean (SD) | 27.7 (24.0) | 31.2 (25.8) | 29.6 (24.8) |
| Median (p25, p75) | 24.1 (3.6, 45.0) | 25.9 (17.3, 31.1) | 25.7 (10.6, 40.3) |
| **Activity monitor: steps taken per day** | | | |
| **Baseline** | N=26 | N=27 | N=53 |
| Mean (SD) | 1755.5 (1963.7) | 1750.9 (1369.6) | 1753.2 (1670.9) |
| Median (p25, p75) | 1292.3 (587.1, 1835.7) | 1572.9 (802.3, 2234.6) | 1448.6 (654.6, 2144.0) |
| **Final** | N=22 | N=25 | N=47 |
| Mean (SD) | 1673.2 (1594.4) | 1836.1 (1647.8) | 1759.9 (1607.4) |

**Table 3** Continued

| | Self-aligning (n=27) | Standard (n=28) | Overall (n=55) |
|---|---|---|---|
| Median (p25, p75) | 1356.4 (204.0, 2719.4) | 1510.3 (922.9, 1908.0) | 1446.3 (551.4, 2270.0) |

TUG and TUDS: faster time indicates better performance; 2mWT: further distance indicates better walking ability; BBS: higher score indicates better balance.

BBS, Berg Balance Scale; 2mWT, 2 min walk test; TUDS, Timed Up and Down Stairs; TUG, Timed Up and Go.

stroke and three falls) were not deemed related to the self-aligning prosthetic ankle-foot. This clinical group is vulnerable to falling, with one recent study stating that over 57% of community-living lower limb prosthesis users fall at least once a year[27] and that 36%–75% of fallers experience recurrent falls.[27 28] The participant who developed blistering on the residuum (the one AE) had been allocated to the intervention group. It is not uncommon for blistering to occur if the skin on the residuum is subjected to increased friction as a result of wearing the prosthesis for longer. In most instances, blisters heal relatively quickly, and the prosthesis user can soon return to normal function with their prosthesis.

### Results in context with other research

A few randomised trials of several different prosthetic ankle-feet (although not self-aligning), evaluating function and self-report measures, have been conducted previously.[29–31] Some of these studies lacked sample size calculations and presented small samples (range n=10–27 participants/trial). Moreover, they represented a heterogeneous group in terms of their mobility level classifications (participants were mostly categorised as 'unlimited community ambulators' (K3) or 'active adults' (K4)), cause of amputation and health comorbidities. The age of the participants in these studies was also younger than in the current feasibility trial (mean range 42.3–57 years). Therefore, the findings from existing studies are difficult to generalise to the majority of people with a lower limb amputation due to dysvascularity, in the UK and other developed nations. We are not aware of any randomised trials investigating prosthetic ankle-feet uniquely involving older adults classified as having 'limited community mobility' as a result of health-related issues.

### Outcome measures

This study was not designed to assess the effectiveness of the self-aligning prosthetic ankle-foot, and no formal comparisons between randomised groups were undertaken as the study was not powered to report effects. Three measures indicated a signal of potential efficacy including the 2mWT (walking ability), daily activity as measured with activity monitors, and quality of life as measured with the EQ-5D-5L. Conversely, other measures (including some population-specific measures and common clinical assessments) did not indicate any difference.

We demonstrated that the use of activity monitors, and their implementation using postal delivery, was feasible in this patient group. We believe this was supported by our study procedures, which included regular participant information letters alerting them to the next stages of the trial, and a detailed bespoke leaflet explaining how participants were requested to fit the activity monitor onto the prosthesis themselves. As the monitor was affixed to the prosthesis, there were few issues with discomfort or discontinued use. Trials using subjective outcomes, where the outcome measure is self-reported by a participant or assessment of an event requires exercise of judgement by an observer, may overestimate the treatment effect.[32] Therefore, it was important to establish the feasibility of using this more objective outcome in a future trial.

On average, both groups took fewer than 1800 daily steps over 30 min throughout the day. The data related to daily steps and stepping time reinforced the largely sedentary behaviour experienced by this clinical group. Our findings are broadly similar to a previous study that reported an average of 1450 (SD 1309) steps/day in a group of older individuals (age: 64 (SD 9) years) with a predominantly transtibial amputation (95% of total participants) as a result of vascular disease when measured over 10 days.[33] Further research is warranted to ascertain whether a more functional prosthetic ankle-foot, compared with current standard care with a non-self-aligning ankle-foot, has the long-term potential to

**Table 4** EQ-5D-5L mean (SD) utility scores, unadjusted mean difference (95% CI) and adjusted for baseline utility at each time point (baseline, interim and final follow-up), by trial arm

| | Self-aligning (n=27) | Standard (n=28) | Unadjusted difference (self-aligning vs standard) (95% CI) | Adjusted difference (self-aligning vs standard) (95% CI) |
|---|---|---|---|---|
| **EQ-5D-5L (average completion 93%)** | | | | |
| **Baseline** | 0.62 (0.30) | 0.63 (0.35) | −0.009 (−0.188 to 0.169) | −0.009 (−0.188 to 0.169) |
| **Interim** | 0.64 (0.18) | 0.57 (0.24) | 0.067 (−0.055 to 0.189) | 0.104 (0.001 to 0.207) |
| **Final** | 0.75 (0.16) | 0.63 (0.32) | 0.120 (−0.021 to 0.262) | 0.117 (0.017 to 0.216) |

increase daily stepping, which could have important implications for patient benefit.

## Health economics

It was feasible to collect the data required for a full-scale trial economic evaluation from this patient population. The high levels of completion suggest acceptability of instruments. The EQ-5D-5L is likely to be sensitive to changes in health states over time. Some changes to the bespoke health resource use questionnaire could be made in future. Use of privately funded healthcare services was negligible so this section could potentially be omitted in a full-scale trial. Participant uptake of the medications in the resource use questionnaire was high and there were some differences between groups suggesting a microcosting exercise including medications data may be merited in a full-scale trial. Although no resource use of walking aids, adaptations and accessories had an uptake greater than 20%, participants were established prosthesis users who may have already accessed the required adaptations prior to entering the trial. Therefore, if a future trial had broader eligibility criteria to include unestablished prosthesis users, it may be important to retain this item.

The approximate NHS cost of the intervention is ~£900 above current standard care as it is assumed that the self-aligning prosthetic ankle-foot is delivered in the usual NHS setting and by the usual staff. There are various self-aligning prosthetic ankle-feet available commercially (eg, using hydraulic mechanisms vs micro-processor controlled and with actively powered propulsion), at varying costs and suitability (eg, some prosthetic ankle-feet are heavier and more suitable for different patient needs). At the present time, we have only considered a single prosthetic ankle-foot, but one that is designed for our clinical population of interest. A full health economics evaluation, as part of full-scale trial, would be able to evaluate whether the initial higher cost of the ankle-foot could be offset by reduced use of other health resources (eg, medications, prosthetic adjustments) and/or increased patient quality of life.

## Strengths and weaknesses

Prosthetics services are an under-researched area within the NHS. As this was one of few clinical trials conducted in this area in the UK, we now have a better understanding of the challenges of recruiting in this setting, such as the limited research resources at clinical sites and variation in how sites organise their patient database systems. The recruiting sites in England provided a wide geographical spread (North East, North West, East of England, Midlands and South of England); however, the participants recruited across these sites were homogeneous in their ethnicity (self-reported as White British, Irish or other). In future, we will develop strategies to maximise participation from patients across different ethnic groups and nations in the UK.

The majority of participants were recruited after a member of their clinical team screened the patient database for potentially eligible participants and posted them a study invitation pack. Therefore, the majority of the screening forms (79%) were initially completed over the phone before the face-to-face screening process confirmed their final eligibility. A limitation of this feasibility study was our lack of prior knowledge of the management of patient databases, which varied across the sites. Four sites used an electronic database for patient records while one used a paper-based system only; one of these sites used three different patient record systems. Not all patient records were up to date, resulting in two participants being randomised in error because they did not fulfil the inclusion criteria fully, and they were subsequently withdrawn. In a future trial, it will be essential to gather relevant information about the organisation of patient records at the local participating sites prior to opening that site to facilitate effective recruitment. This would also help sites recruit participants more consistently across the recruitment period. We were concerned at the outset about the risk of 'resentful demoralisation' among participants allocated to usual care and were mindful of this in developing the patient information for the study. The fact that adherence to the study processes and completion of outcome measures were similar across groups provides some reassurance in this regard. However, this would need continuing attention in the design of a full-scale trial.

## Future research

Based on the findings from this feasibility study, we propose some adaptations to the study protocol for a future full-scale trial. They include:

► Broadening the participant eligibility criteria to include new patients with a transtibial amputation, who would be categorised as having 'limited community mobility' by their multidisciplinary team, and therefore, likely to receive a non-self-aligning prosthetic ankle-foot.
► Lengthening the intervention period of accommodating to the self-aligning ankle-foot, and including a longer follow-up period, would be important for unestablished prosthesis users if the criteria were more inclusive in a full-scale trial.
► Refining the screening process to screen more potentially eligible participants, even if some prove ineligible afterwards, with advanced information about site patient databases.
► Reviewing the secondary outcome measures.
► Redesigning some aspects of the health resource use questionnaire.
► Approaching other prosthetics centres nationally to explore their willingness to be involved in a full-scale trial.

## CONCLUSION

Although we did not reach our target sample, due to slower than anticipated recruitment, 71% of expected participants were in the final analysis, owing to our

retention rate of 93%. Therefore, we believe the STEP-FORWARD study demonstrated feasibility to recruit and retain participants and the ability to obtain a variety of complete datasets collected in clinic and via post. We have also evidenced the ability to deliver the trial across multiple prosthetics centres in England. The knowledge gained and lessons learnt can be implemented in a future definitive trial to determine the effectiveness and cost-effectiveness of a self-aligning prosthetic ankle-foot compared with a non-self-aligning version for people with a transtibial amputation and categorised as having 'limited community mobility'.

**Author affiliations**
[1]Department of Sport, Health and Exercise Science, University of Hull, Hull, UK
[2]York Trials Unit, Department of Health Sciences, University of York, York, UK
[3]School of Science and Technology, Nottingham Trent University, Nottingham, UK
[4]School of Health and Society, University of Salford, Manchester, UK
[5]Specialist Mobility Rehabilitation Centre, Lancashire Teaching Hospitals NHS Foundation Trust, Preston, UK
[6]Disability Medicine and Rehabilitation Unit, Hull University Teaching Hospitals NHS Trust, Hull, UK
[7]Public Involvement Member, UK, UK

**Acknowledgements** This work uses data provided by patients and collected by researchers and members of the NHS who worked collaboratively on the STEPFORWARD trial. The collection of data would not have been possible without the support of the participating NHS Trusts and their staff. The authors would also like to thank the members of the Project Advisory Group (PAG): Dennis Harrison, Peter Wignall, Anthony Gick and Kevin Waller for their invaluable advice on many aspects of the project.

**Contributors** NV led on the conception, design and writing of this manuscript. NM, EC, JW, KB, CM, CB, MT, FJ, AS and DH contributed to the design, writing, critical review of intellectual content and final manuscript approval. All authors agree to be accountable for their work. As chief investigator, NV takes overall responsibility for the work. EC provided statistical expertise in the study design and development stages of the project and the protocol. NM, JW and CM made substantial contributions to the trial design and management. DH provided particular input to PPI. KB was specifically responsible for the health economic aspects of the study design and CB, MT, FJ and AS were responsible for providing expert clinical support.

**Funding** This paper presents independent research funded by the National Institute for Health Research (NIHR) under its Research for Patient Benefit (RfPB) Programme (Grant Reference Number PB-PG-0816-20029). Hull University Teaching Hospitals NHS Trust are the sponsors of the study, on behalf of the funder. Blatchford (Basingstoke, UK) provided the prosthetic feet examined in this research at no cost. The prosthetic feet are currently commercially available devices. Blatchford have had no part in the study design nor are they be involved directly in the research, its subsequent analysis and ultimate dissemination.

**Disclaimer** The views expressed are those of the author(s) and not necessarily those of the NHS, the NIHR, the Department of Health and Social Care, or Blatchford.

**Competing interests** None declared.

**Patient and public involvement** Patients and/or the public were involved in the design, or conduct, or reporting, or dissemination plans of this research. Refer to the Methods section for further details.

**Patient consent for publication** Not required.

**Ethics approval** The trial received favourable ethical approval from the National Health Service (NHS) Yorkshire and the Humber—Leeds West Research Ethics Committee (ref:18/YH/0089) and Health Research Authority in May 2018.

**Provenance and peer review** Not commissioned; externally peer reviewed.

**Data availability statement** Data are available on reasonable request. The data that support the findings of this study are available from the corresponding author, (NV), on reasonable request.

**ORCID iDs**
Natalie Vanicek http://orcid.org/0000-0002-9602-3172
Judith Watson http://orcid.org/0000-0003-0694-3854
Catriona McDaid http://orcid.org/0000-0002-3751-7260
Abayomi Salawu http://orcid.org/0000-0003-2496-0826

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
