## [Reviewer comments · BMJ Open]

ARTICLE DETAILS

TITLE (PROVISIONAL)	The STEPFORWARD study: a randomised controlled feasibility trial of a self-aligning prosthetic ankle-foot for older patients with vascular-related amputations
AUTHORS	Vanicek, Natalie; Coleman, Elizabeth; Watson, Judith; Bell, Kerry; McDaid, Catriona; Barnett, Cleveland; Twiste, Martin; Jepson, Fergus; Salawu, Abayomi; Harrison, Dennis; Mitchell, Natasha

VERSION 1 – REVIEW

REVIEWER	Murray E. Maitland University of Washington United States
REVIEW RETURNED	10-Nov-2020

GENERAL COMMENTS	The STEPFORWARD study: a randomised controlled feasibility trial of a self-aligning prosthetic ankle-foot for older patients with vascular-related amputations Vanicek N, et al. The researchers randomized 55 experienced lower extremity prosthetic users to a sagittal plane adaptable foot prosthesis or continued use of the usual prosthesis. Recruitment was found to be slower than expected and they did not hit targets. Outcomes showed little difference between arms. However, completion rates were high and adverse events were low. Suggestions Introduction The authors should take an opportunity to justify relevance to a broad range of readers since this is a more general journal compared to JPO, for example. Lines 24 – 26. “Subsequently, the majority of older people...” The term uniaxial and multiaxial should be defined because the readership of this journal is not likely to understand the design features of these feet. Lines 26- 28. “These prostheses cannot align...” This sentence and the next need citations. The second of these sentences can be supported by published literature. Lines 26- 28. “These prostheses cannot align...” Since the concept behind a multiaxial foot is that it has limited adaptation to uneven and sloped ground, the authors might need to explain the limitations in this case. Lines 37 – 60. “A self-aligning prosthetic ankle-foot...” This reader cannot follow the characteristics of the prosthetic foot. It was very
--

	difficult to determine which foot the authors were describing until the methods. The first citation has four feet in the study. If the authors are describing the Avalon or a family of feet, then just say it. Methods Participants. Include exclusion criteria. The authors state there are two levels of screening. There should be enough information in the paper to understand these two levels, and for people to reproduce them. Intervention. This reviewer finds the participant's outcome unethical. The outcome may also bias users. One group of participants is offered a new prosthetic foot, but only if they prefer it. The other group is not offered anything but the same old foot. The compensation is therefore very different between the groups. The intervention group was also offered free prosthetics services. Were other incentives offered? The authors should be clear to readers about differences in benefits and how this might affect the outcomes. The authors suggest that a feasibility study is needed to justify a randomized controlled trial. However, the researchers have recruited potential participants into an incomplete study. Does this mean there will be challenges in the future, because of this previous recruitment? Or, can this information be extrapolated in a valid manner? Results. Identification of the primary outcome measure. Earlier in the paper, the authors state that they will present a sample size as evidence of the feasibility. The authors should present information in the results regarding sample size calculations. Somewhere, there should be a statement about which outcomes would, or would not be viable. Discussion. Main findings. The authors should present challenges as well as the strengths of the approach. Only 7% of potential participants were deemed to be eligible. Recruitment was slower than expected. The number of participants per location was important. The information regarding challenges recruiting people from different ethnic groups is excellent. "The majority of patients were recruited..." This is excellent information. Results in the context of other research. The current project had 55 participants. From a prosthetics trial perspective, that is quite large. The researchers should describe what they mean by a "full-scale RCT". The researchers should discuss the implications of outcome results. For example, the discussion includes the feasibility of using activity monitors but not the reality of participants' steps being reduced over time in one arm compared to the other. Are the outcomes valid for this research design? The conclusion here and in the abstract should be clear about the challenges, not just "feasible". Is there a chance this will be a conclusive trial given the outcomes and sample size?
--	---

	Edits Introduction. Lines 4-6. “ In the UK, ...” This is a run-on sentence that makes it awkward. Dividing it up into the incidence of amputations and the characteristics of the people with amputations would improve readability. Lines 7-9. “Major lower limb amputations...” The term “coronary heart” is redundant. The authors might mean “coronary” or “coronary arterial”. Lines 7-9. “Major lower limb amputations...” The authors might be referring to comorbidities and not causes of lower extremity amputation. This reader is not clear how cerebrovascular disease causes lower extremity amputation, for example. Line 12 “...categorized as “limited mobility.” This reviewer suggests “...categorized as having limited community mobility.” Readers may wish the quoted term defined. Lines 51-55. “Although a self-aligning ankle-foot is more expensive...” There are several sentences in this paper that are challenging for readability. Having “although” twice creates too many subordinate clauses. The entire bracketed phrase could be deleted. Another example is directly before this sentence. Functional prostheses should improve function, but there is limited information about what this means. The authors should consider an editor to improve readability. Methods Line 11 “The self-aligning ankle-foot...” The webpage says AvalonK2, without the K2 in superscript. References In-line citations should not have brackets. Reference #1, #23. As a reader, giving the document URL would be helpful. This might be up to the editor. Journal names should be PubMed abbreviations. For example, reference #3 should be “Health Technol Assess”. Please check throughout. #8 A book title is typically capitalized. Check the information for authors. #16, #21, #24, #26 The title of a journal paper is not typically capitalized. #20 This should be “Two-, six-, and 12-minute walking tests in respiratory disease”. Table 1. Cause of amputation. Several of these proposed causes relate back to diabetes. The categories are not independent. The authors should be clear to readers how one category was determined to be the correct category.
--	--

REVIEWER	Max Shepherd Georgia Tech, USA
REVIEW RETURNED	14-Nov-2020

GENERAL COMMENTS	This could be a helpful study for determining if new hydraulic ankles can improve quality of life for amputees. Early evidence from a number of sources is promising, and a randomized controlled trial can help determine actual quality of life improvements, if they exist, in people with limited mobility, who represent the majority of amputations but are underrepresented in most research. I have only
--

	a couple minor comments: Page 9, Lines 37-38 " For the clinical assessments, participants with the self-aligning prosthetic anklefoot walked on average 7.8 metres further at final follow-up compared to baseline." Please include mention of the test. Also consider adding either the total metres walked in each condition, or report it also as percentage, so the size of the effect can be more readily understood. Also consider adding standard deviations reported in your results section. The "minutes standing per day" values do not make sense to me. Is it possible these were the aggregate for the whole week? I am reading your results as the average subject stands for ~16 hours / day... Please either correct or clarify. This may be because it's just a draft, but please ensure that Figure 1 is included in the final submission as a vector image instead of a bitmap, as it is very difficult to read as a bitmap image.
--	--

VERSION 1 – AUTHOR RESPONSE

Reviewer: 1

Reviewer Name: Murray E. Maitland

Institution and Country: University of Washington, United States Please state any competing interests or state 'None declared': None declared

The researchers randomized 55 experienced lower extremity prosthetic users to a sagittal plane adaptable foot prosthesis or continued use of the usual prosthesis. Recruitment was found to be slower than expected and they did not hit targets. Outcomes showed little difference between arms. However, completion rates were high and adverse events were low. **Thank you for your comments on our paper. All corrections are marked in red in the manuscript.**

Suggestions

Introduction

The authors should take an opportunity to justify relevance to a broad range of readers since this is a more general journal compared to JPO, for example. **We have included a sentence stating the range of professionals who might find this manuscript relevant. We believe the article meets the BMJ Open remit of publishing across a range of medical disciplines and therapeutic areas.**

Lines 24 – 26. "Subsequently, the majority of older people..." The term uniaxial and multiaxial should be defined because the readership of this journal is not likely to understand the design features of these feet. **We have explained the design features of the uniaxial and multiaxial feet**

Lines 26- 28. "These prostheses cannot align..." This sentence and the next need citations. The second of these sentences can be supported by published literature. **We have added citations as suggested.**

Lines 26- 28. "These prostheses cannot align..." Since the concept behind a multiaxial foot is that it has limited adaptation to uneven and sloped ground, the authors might need to explain the limitations in this case. **We have expanded this section to explain that the lack of functionality of 'rigid' feet is related to compensatory actions required to achieve walking stability.**

Lines 37 – 60. “A self-aligning prosthetic ankle-foot...” This reader cannot follow the characteristics of the prosthetic foot. It was very difficult to determine which foot the authors were describing until the methods. The first citation has four feet in the study. If the authors are describing the Avalon or a family of feet, then just say it. Thank you for drawing this lack of clarity to our attention. In this paragraph we are describing a ‘family’ of ankle-feet that use a hydraulic mechanism to self-align to sloped surfaces. We have made some modifications to this paragraph to clarify we are referring to a ‘family’ of feet and have specified they are designed for active users and also for users having ‘limited community mobility’. We don’t want to specify any particular type of foot or manufacturer at this stage.

Methods

Participants. Include exclusion criteria. The authors state there are two levels of screening. There should be enough information in the paper to understand these two levels, and for people to reproduce them. We are conscious that the paper is already quite long. We have previously published the full protocol for the study and it is available open access so we have only provided a summary of the methods. We have added some further details to the section on ‘Recruitment and Randomisation’ and also referenced the protocol in this section to draw the reader’s attention to where further details can be found. We have also added the exclusion criteria. We agree they are important to balance the inclusion criteria.

Intervention. This reviewer finds the participant’s outcome unethical. The outcome may also bias users. One group of participants is offered a new prosthetic foot, but only if they prefer it. The other group is not offered anything but the same old foot. The compensation is therefore very different between the groups. The intervention group was also offered free prosthetics services. Were other incentives offered? The authors should be clear to readers about differences in benefits and how this might affect the outcomes. Our study protocol was reviewed and approved by the funder (National Institute of Health Research (NIHR)) and by the National Health Service Yorkshire and The Humber – Leeds West Research Ethics Committee. This included approval for patient-facing information explaining the study and the two alternatives to which they would be randomised. A Patient Advisory Group was also involved in designing the materials. To clarify, to be included in the study, participants agreed to be randomised to either group. The intervention group received the self-aligning prosthetic ankle-foot and they were asked to use it throughout the study intervention period. They were not asked for their preference to use the self-aligning foot at the time of group allocation. At the end of the trial, they were given the option of keeping the self-aligning foot or reverting back to their normal non-self-aligning foot. As the device is classified as a medical device, it could not be given to anyone else. All participants were NHS patients and all prosthetic services were free to both groups. The only difference in care was that one group received a different prosthetic ankle-foot. We have included a sentence (which aligns with our published protocol paper) to clarify this point in the revised manuscript. Finally, when potential participants were deciding whether to consent to taking part, the following wording in the *Participant information sheet* made it very clear on whether or not they would be able to use the self-aligning foot:

12. Will I get to trial the new prosthesis if I’m in the standard treatment group?

At this stage, we don’t know that the new prosthesis is better and more acceptable to patients than the standard prosthesis. That is why we are doing this research. It is not common practice to allow patients to sample a new prosthesis. Until we are confident that the new prosthesis is more effective, we are unable to let participants in the standard treatment group try it out.

13. Will I get to keep the new prosthesis if I’m in the novel treatment group?

Participants in the novel treatment group will be able to keep the new prosthesis if they preferred it.

We agree there is a risk of bias in an unblinded study and we were concerned at the outset at the risk for ‘resentful demoralisation’ amongst participants allocated to the control group. This would have manifested itself in poor compliance in study processes and conversely higher compliance amongst those allocated to the self-aligning prosthetic ankle-foot. However, completion of outcome measures and loss to follow-up were similar across both groups. We have added this information to the discussion.

The authors suggest that a feasibility study is needed to justify a randomized controlled trial. However, the researchers have recruited potential participants into an incomplete study. Does this mean there will be challenges in the future, because of this previous recruitment? Or, can this

information be extrapolated in a valid manner? We conducted a feasibility study for several reasons. In the UK, there have been few clinical trials involving prosthetics therapies and the research infrastructure in prosthetics centres varies widely. If this smaller, cheaper feasibility study proved unacceptable to participants or unable to recruit, then a larger, more expensive trial would result in more time and money wasted. There is much to learn from conducting a smaller study first to inform a larger study.

Although our feasibility trial did not reach its planned sample size of 90, we would not consider it to be incomplete. It meets the design and reporting criteria for a feasibility trial (please see reference #15 in the manuscript: Eldridge SM, Lancaster GA, Campbell MJ, Thabane L, Hopewell S, Coleman CL, et al. Defining feasibility and pilot studies in preparation for randomised controlled trials: Development of a conceptual framework. PLoS ONE. 2016;11(3):e0150205). The aim of a feasibility trial is to inform on the feasibility of a larger scale trial, and that has been achieved here. Even though we under recruited, the number of participants in our study still allowed us to examine the response rates, amount of missing data, and gave indication to the primary outcome of best choice. These will all inform future design including sample size. We additionally learnt a lot from the lack of recruitment to this trial. There may always be some challenges to recruitment, some which may be outside the research team's control. However, in a future trial, our intention is to widen the eligibility criteria and to recruit from more centres. We have added a statement in 'Future research' to explain we would approach other centres for involvement in the full-scale trial and also clarified in the Abstract that amendments would be needed to a future study.

Results. Identification of the primary outcome measure. Earlier in the paper, the authors state that they will present a sample size as evidence of the feasibility. The authors should present information in the results regarding sample size calculations. Somewhere, there should be a statement about which outcomes would, or would not be viable. We have explained in the results that three outcome measures showed a signal of efficacy and that these would be explored with our Patient advisory group to inform our choice of primary outcome in future. This is consistent with our stated objective to "Identify a primary outcome measure(s) for a future main trial". Another published feasibility study also discussed a possible primary outcome measure but did not indicate a sample size (see Hughes E, Mitchell N, Gascoyne S, et al. The RESPECT study: a feasibility randomised controlled trial of a sexual health promotion intervention for people with serious mental illness in community mental health services in the UK. BMC Public Health 20, 1736 (2020)). To avoid any confusion, we have removed the 'earlier' statement you refer to so that the reader is not misled into thinking that we will present a sample size calculation in this manuscript. Removal of: "*We consulted with members of our Patient Advisory Group to identify a suitable primary outcome to inform our sample size calculation for a future full-scale RCT*" in the manuscript.

Discussion. Main findings. The authors should present challenges as well as the strengths of the approach. Only 7% of potential participants were deemed to be eligible. Recruitment was slower than expected. The number of participants per location was important. Given our intention of broadening our inclusion criteria in a future trial (for example, including unestablished prosthesis users with a longer intervention and follow-up period), we feel more patients could be potentially eligible in a future trial. However, we agree it is important to be cautious about this. We have added a statement to the discussion emphasising that a future trial should have an internal pilot to confirm the impact on recruitment of the modifications to the study design.

The information regarding challenges recruiting people from different ethnic groups is excellent. Thank you for this comment.

"The majority of patients were recruited..." This is excellent information. Thank you for this comment.

Results in the context of other research. The current project had 55 participants. From a prosthetics trial perspective, that is quite large. The researchers should describe what they mean by a "full-scale RCT". Although this trial may be large in terms of current prosthetics trials, it is not powered to detect a difference in any outcome between those in the two arms. By a full-scale trial we mean a large-scale RCT that has been appropriately powered to detect a difference in the primary outcome between the two arms (and as such has sample size large enough to answer a research question on intervention effectiveness). We have included a sentence in the Methods to clarify this.

The researchers should discuss the implications of outcome results. For example, the discussion includes the feasibility of using activity monitors but not the reality of participants' steps being reduced over time in one arm compared to the other. Are the outcomes valid for this research design? **As this is a feasibility study, with no sample size calculation and insufficient power, we need to be very cautious about over-interpreting the outcome measures. The variability within the sample was also high. In fact, there may be no statistically significant difference. We entirely understand the interest in the outcome measure results given that the sample is relatively large for a community prosthetics study. However, we want to be cautious and follow the guidelines for feasibility studies.**

The conclusion here and in the abstract should be clear about the challenges, not just “feasible”. Is there a chance this will be a conclusive trial given the outcomes and sample size? **We have amended these sections accordingly to emphasise that the design would need modification in the future. There is no intention currently to use this trial as a conclusive trial as this was not the purpose of the feasibility trial. We have amended the section on the ‘Results in the context of other research’ to make this focus on feasibility clearer.**

Edits

Introduction. Lines 4-6. “ In the UK, ...” This is a run-on sentence that makes it awkward. Dividing it up into the incidence of amputations and the characteristics of the people with amputations would improve readability. **We have amended the sentence to enhance readability.**

Lines 7-9. “Major lower limb amputations...” The term “coronary heart” is redundant. The authors might mean “coronary” or “coronary arterial”. **Thank you for this observation; we have amended the statement accordingly.**

Lines 7-9. “Major lower limb amputations...” The authors might be referring to comorbidities and not causes of lower extremity amputation. This reader is not clear how cerebrovascular disease causes lower extremity amputation, for example. **Thank you for this observation; we have amended the statement accordingly.**

Line 12 “...categorized as “limited mobility.” This reviewer suggests “...categorized as having limited community mobility.” Readers may wish the quoted term defined. **Throughout the manuscript, we have amended the terminology to: having ‘limited community mobility’.**

Lines 51-55. “Although a self-aligning ankle-foot is more expensive...” There are several sentences in this paper that are challenging for readability. Having “although” twice creates too many subordinate clauses. The entire bracketed phrase could be deleted. **We have reworded this sentence removing the use of ‘although’ twice. We have also amended other sentences within the script to improve readability.**

Another example is directly before this sentence. Functional prostheses should improve function, but there is limited information about what this means. **We agree that they should, but we cannot claim they do without robust RCTs. We have elaborated on this in the manuscript.**

The authors should consider an editor to improve readability. **Thank you for this suggestion. We have edited the manuscript to improve readability.**

Methods Line 11 “The self-aligning ankle-foot...” The webpage says AvalonK2, without the K2 in superscript. **Yes, you’re right, it does, but the brochure and other documents have the K2 superscript.**

References

In-line citations should not have brackets. **Thank you. We have made the necessary corrections.**

Reference #1, #23. As a reader, giving the document URL would be helpful. This might be up to the editor. **Thank you. We will seek the editor/publisher guidelines on this.**

Journal names should be PubMed abbreviations. For example, reference #3 should be “Health Technol Assess”. Please check throughout. **Thank you. We have made the necessary corrections.**

#8 A book title is typically capitalized. Check the information for authors. **This was in fact an incorrect reference, which we have now amended.**

#16, #21, #24, #26 The title of a journal paper is not typically capitalized. Thank you. We have made the necessary corrections.

#20 This should be "Two-, six-, and 12-minute walking tests in respiratory disease". Thank you. We have made the necessary corrections.

Table 1. Cause of amputation. Several of these proposed causes relate back to diabetes. The categories are not independent. The authors should be clear to readers how one category was determined to be the correct category.

The related cause of amputation was self-reported by the participant. They could select multiple options if applicable – the footnote indicating this was accidentally omitted from the table, but has now been re-included. There was no further verification of cause of amputation.

Reviewer: 2

Reviewer Name: Max Shepherd

Institution and Country: Georgia Tech, USA Please state any competing interests or state 'None declared': None declared.

This could be a helpful study for determining if new hydraulic ankles can improve quality of life for amputees. Early evidence from a number of sources is promising, and a randomized controlled trial can help determine actual quality of life improvements, if they exist, in people with limited mobility, who represent the majority of amputations but are underrepresented in most research. Thank you for your positive feedback. I have only a couple minor comments:

Page 9, Lines 37-38 " For the clinical assessments, participants with the self-aligning prosthetic ankle foot walked on average 7.8 metres further at final follow-up compared to baseline." Please include mention of the test. Thank you, we have made these changes. Also consider adding either the total metres walked in each condition, or report it also as percentage, so the size of the effect can be more readily understood. Thank you, we have made these changes. Also consider adding standard deviations reported in your results section. We have made these changes.

The "minutes standing per day" values do not make sense to me. Is it possible these were the aggregate for the whole week? I am reading your results as the average subject stands for ~16 hours / day... Please either correct or clarify. In retrospect, we have decided to omit the data on time spent standing and sitting to avoid confusion. Our protocol stated that: "The activity monitor will quantify the time spent during walking activities and the number of daily steps taken". We have therefore presented these data in-line with your previous comment related to reporting the total daily steps/time spent waking with standard deviations in the results section.

This may be because it's just a draft, but please ensure that Figure 1 is included in the final submission as a vector image instead of a bitmap, as it is very difficult to read as a bitmap image. Thank you, we will check this with the editor/publisher.

VERSION 2 – REVIEW

REVIEWER	Murray Maitland The University of Washington, the United States of America
REVIEW RETURNED	17-Dec-2020

GENERAL COMMENTS	The objective, as stated in the abstract includes cost-effectiveness. Therefore, the abstract should contain information related to this outcome. In the methods, the researchers allude to "the primary research question". The research question, "a full-scale RCT" and "fully-powered" need to be defined and evaluated based on the information in the current paper. This reviewer remains concerned that the objective of this study has not been met because of these ambiguities.
--

	The researchers anticipated 72 participants in the final analysis but were only able to include 51. The statement about enlisting other prosthetics centres is insufficient. Since the average recruitment frequency was 1.25 participants per month, what are the logistics of the fully-powered study? Economic analysis. This reader didn't quite understand the cost analysis. The authors should be clear whether the non-self-aligning (usual) foot cost is an estimate or calculated. Are these numbers derived from the person's usual foot? The authors have general remarks regarding outcome measures but stop short of describing what that information implies for an RCT. The authors seem to be saying that the current pilot study does not lead to an efficient choice of outcomes or a suitable number of subjects. "Identification of a primary outcome" was removed from the paper. The researchers describe a study that is a prelude to completing an appropriately powered randomized controlled trial. However, there is no indication at the present time of what that might mean with regard to the number of individuals or the effectiveness of outcome measures in this population. The authors have not been convincing that the current trial is not sufficient to be the full study or that a recruitable sample size will result in a positive trial. The sample size in the current study is relatively large for a study of this type (examples are doi: 10.1016/j.rehab.2018.04.003, 10.1682/JRRD.2014.09.0210, 10.1007/s11999-014-3607-9). The researchers have fairly complete data. They need to demonstrate statistically that a larger trial has the possibility of showing a different conclusion. Grant reviewers at the next level of funding should have this information so they can determine if there will be additional information from a larger trial. Discussion, Results in context with other research. As the authors noted there are very few clinical trials of people using different prosthetic feet. The authors should summarize the strengths and weaknesses of those that do exist either in this section or in the introduction. For example, DOI 10.1682/jrrd.2011.04.0077, 0.1016/j.jbiomech.2014.10.002 and others. Edits List of authors, Barnett, Cleveland "Tecnology" should be "Technology" Twiste, Martin. Does not have a department. Harrison, Dennis, does not have an affiliation Article summary Point 1. This is a clinical trial comparing foot prostheses, not prosthetics services. Introduction, Sentence 3, "Major lower limb amputations..." This sentence is not quite correct. Maybe say that the majority of lower limb amputations have these causes. Methods, Outcome Measures, "The primary outcome..." This reader finds this sentence sounds incomplete. Also, data is plural. Instead of "related to", would it work to say "The primary outcomes were feasibility data..." Results, data completeness, "Conversely, the number of steps taken per day decreased by 82.3 metres in the intervention group..." The
--	---

	authors should make the data and units consistent with the stated outcome.
REVIEWER	Max Shepherd Georgia Institute of Technology, USA I am an author on a patent for a different type of semi-active prosthetic ankle than described in this feasibility study.
REVIEW RETURNED	20-Dec-2020
GENERAL COMMENTS	Several minor comments of things I recommend fixing prior to publication: 1) The new description of a recoil mechanism is a bit awkward, but that may be more common terminology for clinical readers. I suggest replacing "recoil mechanism" with "elastic recoil", and potentially to clarify that a spring or visco-elastic element is providing this returning force to a neutral angle that is more appropriate for standing or walking on level ground. 2) I recommend clearly describing what the numbers in parentheses represent in the second paragraph of the data completeness section. Providing SD of the differences observed in repeated measures (for example, the 7.8 m further walked at follow up than baseline) would also help readers interpret the signal-to-noise ratio (e.g., 7.8 m +- 1 m is a very different results than 7.8 m +- 20 m, which would make the effect seem more like noise)

VERSION 2 – AUTHOR RESPONSE

Reviewer: 1

Reviewer Name: Murray E. Maitland

Institution and Country: University of Washington, United States

Please state any competing interests or state 'None declared': None declared

Thank you for your comments on our paper. All corrections are marked in red in the manuscript.

Comments to the Author:

The objective, as stated in the abstract includes cost-effectiveness. Therefore, the abstract should contain information related to this outcome.

**The objective, as stated in the abstract, is to assess the feasibility of a future trial of cost effectiveness, not to assess cost-effectiveness in the feasibility study. We have already included information related to this objective by stating that we collected 'health resource use data with a bespoke questionnaire' and we present the feasibility results from this stating that the average completion rate for the bespoke resource use questionnaire was 63%.

In the methods, the researchers allude to "the primary research question". The research question, "a full-scale RCT" and "fully-powered" need to be defined and evaluated based on the information in the current paper. This reviewer remains concerned that the objective of this study has not been met because of these ambiguities.

**We have included a sentence to clarify the meaning of a full-scale trial.

The researchers anticipated 72 participants in the final analysis but were only able to include 51. The statement about enlisting other prosthetics centres is insufficient. Since the average recruitment

frequency was 1.25 participants per month, what are the logistics of the fully-powered study?

**The lessons learned from this feasibility study indicate the importance of ensuring we approach all prosthetics sites with a view to gauging their interest, capability and capacity in becoming a recruitment site for a future trial prior to beginning the trial. This was included in the section 'Future research'. In our experience full scale RCTs with a recruitment rate of 1.25 participants per site per month can successfully meet target recruitment. Of course, our recruitment strategy will factor in the recruitment rate and recruitment period when determining the minimum number of sites needed for the future trial.

Economic analysis. This reader didn't quite understand the cost analysis. The authors should be clear whether the non-self-aligning (usual) foot cost is an estimate or calculated. Are these numbers derived from the person's usual foot?

**We have now specified that the non-self-aligning foot is an estimated cost.

The authors have general remarks regarding outcome measures but stop short of describing what that information implies for an RCT. The authors seem to be saying that the current pilot study does not lead to an efficient choice of outcomes or a suitable number of subjects. "Identification of a primary outcome" was removed from the paper. The researchers describe a study that is a prelude to completing an appropriately powered randomized controlled trial. However, there is no indication at the present time of what that might mean with regard to the number of individuals or the effectiveness of outcome measures in this population. The authors have not been convincing that the current trial is not sufficient to be the full study or that a recruitable sample size will result in a positive trial. The sample size in the current study is relatively large for a study of this type (examples are doi: 10.1016/j.rehab.2018.04.003, 10.1682/JRRD.2014.09.0210, 10.1007/s11999-014-3607-9). The researchers have fairly complete data. They need to demonstrate statistically that a larger trial has the possibility of showing a different conclusion. Grant reviewers at the next level of funding should have this information so they can determine if there will be additional information from a larger trial.

**The aim of a feasibility study is to ascertain whether a future, large study would be possible to conduct based on participant recruitment, consent, retention and acceptability of study procedures. We have learned a great deal from the current study and will take this new information forward when designing a future study. We have already outlined the implications for a future trial in the section 'Future research'.

As we have explained in our previous response, the current study is a feasibility study. Consequently, it is not powered to detect any statistical differences and we will remain cautious by following the guidelines for feasibility studies.

Please see doi: 10.5014/ajot.2013.006270 "The outcomes of most feasibility and pilot studies should be measured with descriptive statistics, qualitative analysis, and the compilation of basic data related to administrative and physical infrastructure."

And please see the BMJ 2010 CONSORT statement <https://doi.org/10.1136/bmj.i5239> "The primary focus is on methods for dealing with feasibility objectives. These methods are often based on descriptive statistics such as means and percentages but might also be narrative descriptions (example 1). Typically, any estimates of effect using participant outcomes as they are likely to be measured in the future definitive RCT would be reported as estimates with 95% confidence intervals without P values—because pilot trials are not powered for testing hypotheses about effectiveness."

Moreover, our study objectives (which did not include statistical analysis) were reviewed and approved by the National Health Service ethics committee and our funder the National Institute of Health Research. Our Statistical Analysis Plan was approved by external members of our Trial Steering Committee.

The links that you included above were to randomised trials involving prosthetic components with small sample sizes that were not actually powered to detect a difference in any outcome. Our intention is to take a more cautious approach when identifying any differences between trial arms by conducting a future trial that is appropriately powered to detect a difference in our chosen primary outcome. At the next funding stage, we will be in a position to present grant reviewers with convincing evidence (with vital input from our patient advisory group and additional patient public involvement sessions) about our primary outcome and required sample size (0.9 power) to recruit participants over a 24-month period (adjusted for attrition).

Discussion, Results in context with other research. As the authors noted there are very few clinical trials of people using different prosthetic feet. The authors should summarize the strengths and weaknesses of those that do exist either in this section or in the introduction. For example, DOI 10.1682/jrrd.2011.04.0077, 0.1016/j.jbiomech.2014.10.002 and others.

**We are conscious the paper is already considerably over the word count but we have referred to three studies briefly in the Discussion section. We are happy to take the guidance of the editor if further detail is appropriate.

Edits

List of authors,

Barnett, Cleveland “Tecnology” should be “Technology”

**We have corrected this typo

Twiste, Martin. Does not have a department.

**We have added an affiliation

Harrison, Dennis, does not have an affiliation

**Mr Harrison is a patient public involvement member and does not have an academic affiliation

Article summary

Point 1. This is a clinical trial comparing foot prostheses, not prosthetics services.

**We have amended the sentence accordingly.

Introduction, Sentence 3, “Major lower limb amputations...” This sentence is not quite correct. Maybe say that the majority of lower limb amputations have these causes.

**Thank you. We have made this amendment.

Methods, Outcome Measures, “The primary outcome...” This reader finds this sentence sounds incomplete. Also, data is plural. Instead of “related to”, would it work to say “The primary outcomes were feasibility data...”

**We have rephrased this sentence to include a verb.

Results, data completeness, “Conversely, the number of steps taken per day decreased by 82.3 metres in the intervention group...” The authors should make the data and units consistent with the stated outcome.

**Thank you for spotting this. There was an error in the unit of measurement.

Reviewer: 2

Dr. Max Shepherd, Northwestern University

Comments to the Author:

Several minor comments of things I recommend fixing prior to publication:

Thank you for your comments on our paper.

1) The new description of a recoil mechanism is a bit awkward, but that may be more common terminology for clinical readers. I suggest replacing "recoil mechanism" with "elastic recoil", and potentially to clarify that a spring or visco-elastic element is providing this returning force to a neutral angle that is more appropriate for standing or walking on level ground.

**We have amended the sentence as recommended while still maintaining readability.

2) I recommend clearly describing what the numbers in parentheses represent in the second paragraph of the data completeness section.

**We have included this towards the beginning of that paragraph.

Providing SD of the differences observed in repeated measures (for example, the 7.8 m further walked at follow up than baseline) would also help readers interpret the signal-to-noise ratio (e.g., 7.8 m +- 1 m is a very different results than 7.8 m +- 20 m, which would make the effect seem more like noise).

**Thank you for this suggestion. These data can't be found in the Tables so we went back to our dataset to determine the difference between final follow-up and baseline measures and then computed the mean (SD) values for each trial arm. We now present these data in the Results text. We have also done these calculations for the daily steps data so that the data are presented consistently within the same paragraph.

VERSION 3 – REVIEW

REVIEWER	Murray Maitland University of Washington, United States
REVIEW RETURNED	05-Feb-2021

GENERAL COMMENTS	The paper's value is greatly diminished because there is no analysis of the quantitative information. The primary goal of this paper, to demonstrate that the proposed study is doable is not answered. All that the reader is told by the end is that the participants can be enrolled, a part of the overall equation. This reviewer does not agree with the authors' reluctance to be specific about the meaning of "full-scale trial". The authors use words like "powered appropriately" which indicates the current study will lead to a definitive research approach. The authors state that the proposed research will have a sample size. Given the current data, the authors can put this into a logistical framework. Typically, this means that the intervention has an effect size and the outcomes have a meaningful change. As an example, this reviewer is proposing a cross-over design with a sample size of 30; there will be over 80% power for detecting a standardized effect size of 0.52 based on our current study. The authors also discuss the need for primary outcomes that goes unanswered. This is part of the authors' "if so, how" argument. In the results about the "primary outcomes", the authors are vague, stating that the outcomes are feasible. Since the measures have been used for this population previously, it seems that the question has been answered. More importantly, readers are not made aware of whether outcomes have feasibility or clinical potential to show differences with specified sample size. At the present time, this reviewer interprets the data to mean that if a trial were to go ahead, there is no evidence to show that an outcome will succeed in showing a difference regardless of the trial size. Yet the authors
---

	persist in saying that some unspecified “full-scale” RCT will. For example, daily activity with the activity monitor gets worse with the intervention. However, the authors seem to persist in suggesting that this outcome be used. Without the raw data, readers cannot accurately predict the potential for a positive trial. All research study proposals should have this prediction, which is missing here. To be absolutely clear: which outcomes will have the potential for a positive trial, how many participants will it take, and how long (how many locations)??? The authors could provide excellent information to other researchers and could put this into context, but they don’t. Is the proposed research design the most efficient compared to other research designs? The authors do not discuss clinically meaningful differences and the populations needed to determine whether this product meets that goal. A statistician would be able to give some excellent information. A limitation of the current study is that the authors stopped short of a statistical analysis of their goals – developing a fully-powered study design. This reviewer remains concerned that the tone of language in the discussion of activity monitors is misleading compared to data. On page 12, lines 29 to 60, the authors purport that activity monitors show “potential efficacy”. Besides the fact that this is not an efficacy trial, the authors fail to mention that the intervention group went from 30 to 28 minutes of stepping per day, and 1756 to 1673 steps per day. Therefore, the authors’ point that step count could be used “to ascertain whether a more functional prosthetic ankle-foot ... has the long-term potential to increase daily stepping” is contradictory to the facts. Readers should not need to work this hard to understand the implications of the current research. The authors suggest that people with new amputations should be enrolled. This reviewer has not commented previously on this statement, and therefore changes to the manuscript are not expected in this regard. However, this reviewer is curious how the authors believe people with stable prosthesis use or new-onset prosthesis use can be mixed. One might expect a subgroup analysis which would increase the number of participants required. In summary, the reason this reviewer has a statistician included in the pre-application phase of research is specifically for the purpose suggested by the authors of the current paper. This reviewer would enjoy seeing the output of that analysis, and the paper would be much stronger.
--	--

VERSION 3 – AUTHOR RESPONSE

Rev #1: The paper’s value is greatly diminished because there is no analysis of the quantitative information. The primary goal of this paper, to demonstrate that the proposed study is doable is not answered. All that the reader is told by the end is that the participants can be enrolled, a part of the overall equation.

** As we have explained in our previous responses, the current study is a feasibility study. Consequently, it is not powered to detect any statistical differences and therefore it would not be appropriate to conduct a statistical analysis of the quantitative data.

Rev #1: This reviewer does not agree with the authors’ reluctance to be specific about the meaning of

“full-scale trial”. The authors use words like “powered appropriately” which indicates the current study will lead to a definitive research approach. The authors state that the proposed research will have a sample size. Given the current data, the authors can put this into a logistical framework. Typically, this means that the intervention has an effect size and the outcomes have a meaningful change. As an example, this reviewer is proposing a cross-over design with a sample size of 30; there will be over 80% power for detecting a standardized effect size of 0.52 based on our current study. The authors also discuss the need for primary outcomes that goes unanswered. This is part of the authors' “if so, how” argument.

** We have already clearly defined that a full-scale trial will be powered to detect a difference in the primary outcome. This will be informed by further patient and public involvement (PPI) work to finalise the primary outcome and an appropriate minimal clinically important difference will be used.

Rev #1: In the results about the “primary outcomes”, the authors are vague, stating that the outcomes are feasible. Since the measures have been used for this population previously, it seems that the question has been answered. More importantly, readers are not made aware of whether outcomes have feasibility or clinical potential to show differences with specified sample size. At the present time, this reviewer interprets the data to mean that if a trial were to go ahead, there is no evidence to show that an outcome will succeed in showing a difference regardless of the trial size. Yet the authors persist in saying that some unspecified “full-scale” RCT will. For example, daily activity with the activity monitor gets worse with the intervention. However, the authors seem to persist in suggesting that this outcome be used. Without the raw data, readers cannot accurately predict the potential for a positive trial. All research study proposals should have this prediction, which is missing here.

**The outcomes that we chose to report on the feasibility related to data on recruitment, consent and retention as well as completeness of datasets. We believe the reviewer is mistaken that we ‘persist’ in saying the activity monitor outcome will be used. We state that three measures demonstrated a ‘signal of efficacy’. These three measures will be explored with our patient advisory group in future and these three measures will be considered in a future trial.

Rev #1: To be absolutely clear: which outcomes will have the potential for a positive trial, how many participants will it take, and how long (how many locations)??? The authors could provide excellent information to other researchers and could put this into context, but they don't.

Is the proposed research design the most efficient compared to other research designs? The authors do not discuss clinically meaningful differences and the populations needed to determine whether this product meets that goal. A statistician would be able to give some excellent information. A limitation of the current study is that the authors stopped short of a statistical analysis of their goals – developing a fully-powered study design.

** Please see our previous response. We will explore the three measures that demonstrated a ‘signal of efficacy’ in a future trial. It was never our intention to undertake a statistical analysis as outlined in the published trial protocol (<http://dx.doi.org/10.1136/bmjopen-2019-032924>).

Rev #1: This reviewer remains concerned that the tone of language in the discussion of activity monitors is misleading compared to data. On page 12, lines 29 to 60, the authors purport that activity monitors show “potential efficacy”. Besides the fact that this is not an efficacy trial, the authors fail to mention that the intervention group went from 30 to 28 minutes of stepping per day, and 1756 to 1673 steps per day. Therefore, the authors' point that step count could be used “to ascertain whether a more functional prosthetic ankle-foot ... has the long-term potential to increase daily stepping” is contradictory to the facts. Readers should not need to work this hard to understand the implications of the current research.

** The data on steps per day are clearly reported in the paper. One of the objectives of this feasibility study was to simply identify outcomes that could be used in a future trial, and these will be further informed by our patient advisory group. It would be the aim of a future full-scale trial to determine whether a self-aligning prosthetic ankle-foot is actually more effective compared to the non-self-

aligning prosthesis in relation to the chosen primary and secondary outcomes.

Rev #1: The authors suggest that people with new amputations should be enrolled. This reviewer has not commented previously on this statement, and therefore changes to the manuscript are not expected in this regard. However, this reviewer is curious how the authors believe people with stable prosthesis use or new-onset prosthesis use can be mixed. One might expect a subgroup analysis which would increase the number of participants required.

** We may conduct a sub-group analysis depending on the actual participant demographics in a full-scale trial.

Rev #1: In summary, the reason this reviewer has a statistician included in the pre-application phase of research is specifically for the purpose suggested by the authors of the current paper. This reviewer would enjoy seeing the output of that analysis, and the paper would be much stronger.

** Please see our response to this reviewer's first comment.

Reviewer: 1

Competing interests of Reviewer: None Declared